# Association of Acidotolerant Cyanobacteria to Microbial Mats below pH 1 in Acidic Mineral Precipitates in Río Tinto River in Spain

**DOI:** 10.3390/microorganisms12040829

**Published:** 2024-04-19

**Authors:** Felipe Gómez, Nuria Rodríguez, José Antonio Rodríguez-Manfredi, Cristina Escudero, Ignacio Carrasco-Ropero, José M. Martínez, Marco Ferrari, Simone De Angelis, Alessandro Frigeri, Maite Fernández-Sampedro, Ricardo Amils

**Affiliations:** 1Centro de Astrobiología (INTA-CSIC), Carretera de Ajalvir km 4, Torrejón de Ardoz, 28850 Madrid, Spain; 2Centro de Biología Molecular Severo Ochoa (CSIC-UAM), Cantoblanco, 28049 Madrid, Spain; 3Istituto di Astrofisica e Planetologia Spaziali (INAF), via del Fosso del Cavaliere 100, 00133 Rome, Italy

**Keywords:** extremophiles, acidophiles, cyanobacteria, natrojarosite, endolithic ecosystems, earth analogues, astrobiology

## Abstract

This report describes acidic microbial mats containing cyanobacteria that are strongly associated to precipitated minerals in the source area of Río Tinto. Río Tinto (Huelva, Southwestern Spain) is an extreme acidic environment where iron and sulfur cycles play a fundamental role in sustaining the extremely low pH and the high concentration of heavy metals, while maintaining a high level of microbial diversity. These multi-layered mineral deposits are stable all year round and are characterized by a succession of thick greenish-blue and brownish layers mainly composed of natrojarosite. The temperature and absorbance above and below the mineral precipitates were followed and stable conditions were detected inside the mineral precipitates. Different methodologies, scanning and transmission electron microscopy, immunological detection, fluorescence in situ hybridization, and metagenomic analysis were used to describe the biodiversity existing in these microbial mats, demonstrating, for the first time, the existence of acid-tolerant cyanobacteria in a hyperacidic environment of below pH 1. Up to 0.46% of the classified sequences belong to cyanobacterial microorganisms, and 1.47% of the aligned DNA reads belong to the Cyanobacteria clade.

## 1. Introduction

Cyanobacteria is a group of photosynthetic bacteria, some of which can fix nitrogen and live freely (single-celled or filamentous multicellular) or in symbiosis with certain fungi to form lichens and plants [1]. Cyanobacteria are present in a wide range of terrestrial and aquatic environments, such as fresh water, oceans, in temporarily dampened rocks, hot springs, and in environments with low humidity, or even forming endolithic ecosystems in Antarctic rocks [2,3,4,5,6].

The existence of cyanobacteria in acidic environments has been debated for a long time. Although they have been sought in environments far below pH 2, none been reported thus far. Optical microscope identification of some cyanobacterial species has been reported at pH 3 in saline peaty bog lands and wet paddy fields in Kerala (India), associated with soils originating from acid rock drainage [7]. The biodiversity description of the Los Rueldos mine in Asturias (Spain) reported some cyanobacterial sequences at above pH 2 [8]. Other studies reported cyanobacterial presence in acidic waters above pH 2 [9,10], as well as near pH 2 [11]. Nevertheless, several authors assert the impossibility of the existence of members of this microbial phylum in environments below pH 4–5 [12]. 

Little is known about the mechanisms that might hinder the growth of cyanobacteria at low pH. Different authors have tested diverse possibilities, like difficulties in internal pH regulation [13], photosynthetic efficiency [14], defects on solute transport [15], production of energy [16], energy cost for CO_2_ fixation [16], membrane modification affecting permeability [16], but none of them have given a satisfactory explanation. 

Río Tinto is a 92 km long river, located south-west of the Iberian Peninsula, with its origin in the Andévalo area of Huelva province (Spain) and its mouth in the city of Huelva. Río Tinto has an average pH value of 2.3, with several sampling stations below pH 1 [17]. The presence of iron- and sulfur-oxidizing and reducing prokaryotes and their interaction with the Iberian pyritic belt located in the subsurface are responsible for the extreme conditions of acidity (mean pH 2.3) and the high concentration of heavy metals found in its waters (Fe, Cu, Zn, As, Cr). Therefore, it can be classified as an extremely acidic environment. Acidic waters and mineral precipitates in the river were used by ancient civilizations as an indicator for the presence of mineral ores, as the Río Tinto area was the source of metal extraction for Phoenicians and Romans [18]. 

Although, for many years, it was believed that the extreme conditions of Río Tinto were due to mining activities in the area [18,19,20], there have recently been geophysical, geological, and hydrogeological studies that reported the existence of an underground bioreactor [21,22,23,24] in which the metal sulfides of the Iberian Pyrite Belt are the source of microbial energy, feeding the river with acidic waters.

The biodiversity of Río Tinto has been described in different studies in which the presence of both prokaryotes and eukaryotes has been reported [23,25,26]. The only photosynthetic microorganisms described in the river have been eukaryotic algae [26], but the presence of cyanobacteria has not been reported so far.

On the other hand, the participation of cyanobacteria in biomineralization processes has been described through carbonate precipitation processes (carbonatogenesis) from CO_2_ uptake using solar energy through photosynthesis [27]. Those cyanobacterial biomineralization processes have never been associated neither to acidic soils nor acidic mineral precipitates.

In this research, the presence of microorganisms is reported to be operating in a multi-layer acidic mineral deposit associated to acidic water runoff. These multilayer mineral deposits resemble endolithic ecosystems once they have dried out and reached a high degree of hardness. The observed microniches have stable environmental conditions with no seasonal changes and can thus be considered protective ecosystems where microbes can establish their optimal environmental conditions to develop. This study aimed to identify the microorganisms existing in the green-blue layer of this microbial mat. Scanning and transmission electron microscopy, and fluorescence in situ hybridization (CARD-FISH) allowed for the identification of cyanobacteria in the microbial mat, and metagenomics for the cyanobacterial diversity characterization.

## 2. Materials and Methods

### 2.1. Sampling

The sampling campaign was run in May 2010 for 5 days. The sampling area (37.721451° N, 6.551082° W) was located in an acidic water runoff, close to the origin of Río Tinto, over a mining tunnel wall. This runoff deposits mineral precipitates on the wall forming laminated crusts in which three different layers are visible (Figure 1).

Samples (1 cm thick, immersed in runoff water) were collected aseptically from 5 different spots using sterile material (spatulas, Eppendorf tubes, and 12 mL glass vials sealed with septum tap). Samples for metagenomics were frozen in dry ice. Samples for hybridization were fixed in the field with 4% formaldehyde in phosphate-buffered saline (PBS) for 2 h at room temperature and stored at 4 °C until further processing. Samples in glass vials were also sealed with parafilm tape for their transport to the laboratory under aseptic and controlled conditions at 4 °C. Samples were kept at 4 °C until further processing.

### 2.2. pH Measurement

Physicochemical parameters (pH, temperature, redox potential, and conductivity) were measured in situ using a multi-parametric probe, PD 650, Eutech Instruments (resolution 0.01 pH; relative accuracy +/− 0.002 pH). The pH inside the mineral deposits was measured with a microneedle electrode (Unisense Co., Aarhus, Denmark). The pH and temperature measurements were taken twice a day (at noon and 4 pm) at two different points in the sample over the 5 days and for several periods along the sampling year.

### 2.3. Irradiance Records

Irradiance measurements on the surface and within the mineral deposit were carried out during the day using an Ocean Optics VIS-UV USB4000 fiber optic spectrometer. The measuring range of the instrument was from 200 to 850 nm with a resolution of 1.5 nm. To take measurements below the mineral deposit, an Ocean Optics QP400-2-UV/VIS fiber optic, 400 microns in diameter, was used.

### 2.4. X-ray Diffraction (XRD) Analysis

The different layers of the mineral deposit were separated and analyzed by XRD using a Seifert 3003 TT (GE Sensing and Inspection Technologies GmbH, Hürth, Germany) with Cu Kα radiation (*λ* = 1.542 Å). The X-ray generator was set at an acceleration voltage of 40 kV and a filament emission of 40 mA. Samples were scanned between 5° (2*θ*) and 70° (2*θ*) using a step size of 0.05° (2*θ*) and a count time of 2 s. Data were collected using SCANX and viewed using Analyze (GE Sensing and Inspection Technologies GmbH). Samples were powdered, and the powder was packed into a standard aluminum sample holder. Samples were measured in the same way as the calibration samples. 

### 2.5. Scanning Electron Microscopy (SEM)

Samples were fixed in 4% glutaraldehyde in 0.1 M phosphate buffer (pH 7.2) for 2 h at room temperature, then dehydrated in increasing concentrations of ethanol (30, 50, and 70%) for 20 min each. Critical point drying was applied to the samples (Critical Point Dryer Ball-Tec CPD 030). After drying, the samples were deposited onto conductive graphite stubs. The samples were sputtered and coated with gold in a Bio-Rad SC 502 instrument to prevent charging under the electron beam. The samples were then observed with a Scanning Electron Microscope JEOL JSM-5600 LV and an Electronic microscope XL30S FEG Philips, Netherlands (SEM-FI) with an acceleration voltage of 20 kV and a working distance of 20 mm. Energy-dispersive X-ray spectroscopy (EDX) microanalysis using an INCAx-sight with a Si-Li detector (Oxford, UK) was used for qualitative elemental composition determination, (detection limit of 10% of the main element). Lighter elements (C, O, and N) were detected by the INCAx-sight with a Si-Li detector of the instrument, and the quantitative numerical data of the spectra obtained were referenced as default to the higher peaks obtained in each spectrum (C in our case).

### 2.6. Transmission Electron Microscopy (TEM) and Immunocytochemistry

Small sections, 1 mm, of the different layers (differentiated zones) along the entire mineral deposit of the wall were fixed in 4% paraformaldehyde and 2% glutaraldehyde in phosphate buffer, pH 7.4, for two hours at RT under stirring and overnight at 4 °C. Following this, the sections were extensively washed in the same buffer. Samples were then post-fixed with 1% OsO_4_ in water for 60 min at room temperature in the dark, washed three times with bidistilled water, and incubated with 0.15% tannic acid in buffer for 1 min at room temperature. The samples were then washed with buffer and bidistilled water and incubated with 2% aqueous uranyl acetate for 1 h at room temperature and then washed again. Dehydration was performed in increasing concentrations of ethanol (30, 50, and 70%) for 20 min each, 90% twice for 20 min each, and 100% twice for 30 min each at room temperature. Dehydration was completed with a mixture of ethanol/propylene oxide (1:1) for 10 min and pure propylene oxide for 3 × 10 min. Infiltration of the resin was accomplished with propylene oxide/Epon (1:1) for 45 min and overnight in pure resin, while agitating the samples on a wheel rotator. The next day, the resin was renewed for 2–3 h and the samples encapsulated in BEEM capsules. The resin was subsequently polymerized at 60 °C for two days. Ultrathin sections of 70–80 nm were cut with an ultramicrotome Ultracut E (Leica, Wetzlar, Germany) and mounted on Formvar–carbon-coated Cu/Pd 100 mesh grids prior to being stained with 2% aqueous uranyl acetate and Reynolds’ lead citrate. Samples were examined at 80 kV in a JEM1010 transmission electron microscope (Jeol, Tokyo, Japan) and images were taken with a digital camera, TemCam-F416 (TVIPS, Gauting, Germany).

TEM immunolocalization of the cyanobacterial α-major carboxysome shell protein was conducted with colloidal gold-conjugated specific antibodies on these ultrathin sections mounted on grids. Grids were placed on a drop of 50 mM glycine in T/TBS (Tris-buffered saline, containing 20 mM Tris-HCl pH 7.5, 150 mM NaCl, and 0.3% Tween 20) for 10 min at room temperature for aldehyde quenching. The blocking step was carried out as follows: 5% Bovine serum albumin (BSA) in T/TBS for 1 h at 37 °C. Grids were then floated on 10 μL of diluted anti α-major carboxysome shell protein antibody, 1:2, 1:20, or 1:100 in T/TBS containing 5% BSA and then incubated for 90 min at 37 °C, followed by three washes of 10 min with T/TBS + 0.2% BSA at room temperature. The primary antibody was detected with protein A conjugated to 15 nm colloidal gold particles (PAG15, CMC Utrecht) diluted 1/50 in T/TBS 5% BSA. This incubation was carried out for 1 h at 37 °C. Finally, the grids were subsequently washed with T/TBS + 0.2% BSA, TBS, and bidistilled water at room temperature. The samples, after being post-stained (2% aqueous uranyl acetate and lead citrate) [28], were examined under a JEM-1010 microscope (Jeol, Tokyo, Japan) operated at 80 kV.

### 2.7. Fluorescence “In Situ” Hybridization Combined with Catalyzed Reporter Deposition (CARD-FISH)

Samples were fixed with 4% formaldehyde in phosphate-buffered saline (PBS) for 2 h and stored at 4 °C until further processing. The blue-green layer was separated with sterile spatula and disaggregated in PBS. The samples were concentrated on black membranes of 0.22 μm (Millipore, Darmstadt, Germany), then washed with PBS and absolute ethanol while filtering before the filters were air dried. Once the filters were dried, they were covered with agarose at 0.2%, dried at 37 °C, dehydrated with ethanol, and stored at −20 °C.

CARD-FISH experiments were performed on the membrane filter as previously described [29]. Hybridizations were performed with 5′-HRP-labeled oligonucleotide CYA 361 and EUB 338 I–III probes (Biomers, Ulm, Germany) [30,31,32] and stringency was regulated by adjusting formamide to 35% and NaCl to 80 mM concentration in the hybridization and washing buffers, respectively. An additional inactivation of peroxidases was performed between hybridizations. The filter was counterstained with DAPI (4′,6-diadimino-2-phenylindole) as recommended by the manufacturer, and thereafter mounted on glass slides with a mixture of 1:4 Vectashield (Vector Laboratories, Newark, NJ, USA): Citifluor (Citifluor, London, UK). Microorganism counting (10 fields, above 100 microorganisms/field) was performed on an Axioskop epifluorescence microscope (Zeiss, Jena, Germany). Samples were imaged using a confocal laser scanning microscope LSM710 (Carl Zeiss, Jena, Germany) equipped with a diode (405 nm), argon (458/488/514 nm), and helium and neon (543 and 633 nm) lasers. Finally, the images were processed using Fiji software version 2.9.0 [33]. 

### 2.8. Metagenomics Shotgun Sequencing and Analysis

DNA was extracted from 1 g of sample using the PowerSoil® DNA Isolation Kit (MoBio Laboratories, Carlsbad, CA, USA) as recommended by the manufacturer. DNA was eluted in 50 μL of sterile Milli-Q water and sent for 2 × 250 paired-end sequencing on an Illumina MiSeq v2 platform (Illumina) at the Parque Científico de Madrid Foundation (Madrid, Spain). Reads were quality filtered with the read-qc module of MetaWRAP software version 1.3.2 [34] and assembled with SPAdes v3.13.0 (--meta and --carefull options) [35]. The taxonomic composition of the microbial community was analyzed with Kraken 2 v2.0.9 [36] and visualized with Krona v2.7.1 [37]. Protein-coding gene prediction was carried out with the Prodigal v2.6.3 software [38] and annotation was performed with Diamond v0.9.29 [39], using the NCBI non-redundant (NCBI-nr) protein as the reference database. 

The cyanobacteria-related contig sequences analyzed in this study were submitted to the European Nucleotide Archive (ENA) under the accession number ERZ1300892.

### 2.9. Spectroscopy Measurements Setup

Spectra in the VIS-NIR range were acquired on the field with an ASD FieldSpec4 portable spectrometer in the 0.35–2.5 μm range, with 6 mm spatial resolution and spectral resolution of 3–8 nm. The Spectralon 99 reference standard was used for the measurements, averaging 100 spectra for each acquisition using sunlight as the light source.

Raman spectroscopy was performed at the INAF-IAPS laboratories in Rome (Italy), using a Bruker SENTERRA II Microscope Spectrometer equipped with an Olympus microscope using a 50× long working distance objective. The excitation laser wavelength was 532 nm, and the laser power was kept at the minimum value of 0.5 mW to avoid overheating of the sample.

## 3. Results

### 3.1. Physicochemical Characterization

The thickness of the mineral deposits range from 5 mm and 1 cm wide (Figure 1). These layered structures are deposited over rocks or manmade mining tunnel walls but always in locations with very specific environmental characteristics. They can be found mostly on the shaded side of walls, with very few hours of direct sunlight. The presence of several layers of different colors was clearly observed (Figure 1). The surface layer had a light yellow-brown color with the presence of mineral crystals. The second layer was green-blue, and the third and last layer, mixed very closely with the second green-blue layer, was of an ocher brown color, very characteristic of the mineral precipitates in the surroundings of Río Tinto.

The irradiance spectrum was measured using light microsensors at midday both at the surface and in the blue-green layer (Figure 2). As shown in Figure 2, up to 27% of the irradiance at 650 nm is attenuated in the deepest layer. The incident radiation over the five days of measurement was the same at the same time of the day, with no significant observed changes. 

The pH value at the top and inside the mineral deposits were the same, with a mean of 0.8 (ranging from 0.75 to 0.85, which is a variation of 12.5% according to the differences among the data divided by the absolute value percent), congruent with the mineral composition (see below). Temperature was measured twice a day for five days and we obtained a stable value of 12 °C (ranging from 11.5 to 12.5, which is a variation of 8.33%). The different layers were separated to proceed to mineralogical composition and microscopy analysis (Figure 3).

The mineral composition of the different layers of the mineral deposits was determined by XRD and corresponded mostly to gypsum in the case of the outer layer and natrojarosite to the inner brown ocher layer associated with the green layer (Figure 3). 

### 3.2. Microscopy Observations

Samples from the different layers were analyzed by scanning electron microscopy (SEM). Observation of the green-blue layer rendered images of spherical microorganisms attached to the mineral surface (Figure 4).

The EDX analysis of the mineral surface (arrowheads in Figure 5a,c) indicates a high sulfur and iron content from the natrojarosite (Figure 5b) and a high content of calcium on the surface where the microbial mat adheres (Figure 5d).

These samples were also analyzed by transmission electron microscopy (TEM). The cell morphologies observed were diverse, all spherical and mostly unicellular, some grouped in a consortium of several cells (Figure 6). The diameter of the cells ranged from 0.5 and 1.4 μm.

A high-resolution view of these cells reported an ultrastructure with a cell envelope that is composed of a thick polysaccharide capsule. Outside this envelope, a band of fibrous protein is observed and a mucilaginous layer. Inside, the presence of the cell membranes, the thylakoid-type membranes system, some putative carboxy-type corpuscles, and granules of variable density that could contain glycogen, polyphosphates, or lipid drops (Figure 7) can be observed. 

### 3.3. Immunological and Fluorescence In Situ Hybridization Analysis

To further demonstrate that the cyanobacteria are associated with natrojarosite mineral layers at pH 0.8, immunoassays using α-major carboxysome shell protein antibodies, fluorescence in situ hybridization (CARD-FISH) with a specific probe targeting cyanobacteria, and a metagenomic shotgun to identify the microorganisms present were performed.

The α-major immunolocalization of the cyanobacterial carboxysome shell protein was carried out. TEM observations showed positive immunolocalization, identifying the presence of carboxysomes inside the cells identified as cyanobacteria (Figure 8). 

Double CARD-FISH experiments were performed in membrane filters using the probes CYA 361 and EUB338 I–III, which allowed the detection, quantification, and visualization of the microorganisms belonging to Cyanobacteria and Bacteria, respectively. Positive hybridization clearly shows the presence of cyanobacteria located within a bacterial consortium (Figure 9).

Quantification of microorganisms revealed that 1.93 × 10^11^ microorganisms per gram inhabit the microbial mat. Of those, up to 31.8% of these microorganisms belong to the Bacteria domain and 1.99 × 10^9^ (1.03%) have been identified as members of the phylum Cyanobacteria.

### 3.4. Metagenomic Analysis

Shotgun metagenomics was carried out to characterize the microbial community present in the acidic microbial mat. Taxonomic assignment of the reads indicated that up to 0.46% of the classified sequences belong to cyanobacterial microorganisms. The Synechococcales and Nostocales orders stand out, each comprising 30% of the cyanobacterial metagenome reads, followed by the Oscillatoriales and Chroococcales orders, with 14% and 12% of reads, respectively (Figure 10). 

In addition, taxonomic assignment of open reading frames (ORF) was carried out, indicating that up to 1.47% of the aligned DNA reads belong to Cyanobacteria clade. Among these are the cell components involved in photosynthetic processes, such as the photosystem I iron–sulfur center protein PsaC, with a percentage of identification of 95.1% in a query coverage of 98.8%; the photosystem II D2 protein with 83.5% and 99.0% of identification and coverage, respectively; the allophycocyanin subunit beta with 90.7% of identification with 99.4% of coverage, whose closest related organisms belong to the Pseudanabaenales, Chroococcaceae, and Microcystaceae cyanobacterial groups among others (Appendix A).

In addition, different cyanobacterial ribosomal proteins were identified, such as L34 from *Crinalium epipsammum* with a coverage of 59.1% and identity similarity of 95.7%, and S6 from *Acaryochloris marina* with 35.9% coverage and 88.5% similarity. Other identified coding sequences include the *Synechococcus* sp. PCC 6312 redox-regulated ATPase YchF with a query coverage of 58.2% and identity of 84.3%, *Microcoleus* sp. PCC 71132Fe-2S iron–sulfur cluster binding domain-containing protein with a coverage percentage of 99% and 78.8% identity percentage, and the *Gloeocapsa* sp. PCC 7428 peroxiredoxin (antioxidant activity) with a coverage of 54% and 70.1% of similarity (Appendix A).

### 3.5. VIS-NIR and Micro-Raman spectroscopy

VIS-NIR measurements were performed at three points on the area in which the microbial mats were found. Specific sections of the tunnel walls are found to be covered with salt encrustations of different colors, and there are orange and white encrustations forming on the surface, below which is a green layer due to the presence of cyanobacteria (Figure 11a).

Figure 11b shows the 400–700 nm spectra collected on the three crust types (white, orange, and green) found on the tunnel wall. The spectrum encompasses only the visible range, since the poor illumination conditions inside the tunnel caused the NIR range to be too noisy. The spectrum collected on the orange crust (black spectrum in Figure 11b) shows strong absorption in the region of 400–500 nm. The spectrum collected on the white crust (red spectrum in Figure 11b) has a higher level of reflectance and less pronounced absorption in the region around 500 nm; however, it shows absorption around 430 nm. In contrast, the spectrum collected on the green zone does not show a red slope in the visible range but does show absorption at 440, 620, and 675 nm, a fluorescence peak at 650 nm, and a scattering peak in the green region (532 nm). A sample of the green layer was measured in the laboratory by micro-Raman spectroscopy. The collected spectrum is shown in Figure 11c with Raman peaks at 1522, 1158, and 1106 cm^−1^.

## 4. Discussion

This work reports a photosynthetic ecosystem with the presence of cyanobacteria developing under the cover of a mineral precipitate composed of natrojarosite and at pH 0.8. The place where the microbial mats containing cyanobacteria were detected is located in an area protected from the direct solar radiation exposure and bathed with the acidic waters of Río Tinto at pH 0.8. As microbial mats are included inside mineral precipitates, it is a protected environment. It seems to be an optimal place for the development of cyanobacteria since it has low variations in both temperature and solar irradiance. There are few hours of direct solar incidence, but the site is well illuminated by indirect solar radiation. The photosynthetically active range of light is suitable for the development of photosynthetic organisms. These stable environmental conditions provide the appropriate environment for the development of the green-blue layer inside of the microbial mat. Unfortunately, the thickness of the greenish-blue layer does not allow for the measurement of pHs across the cyanobacterial mat to check for any internal change in pH.

The presence of cyanobacteria was demonstrated using a multi-methodological approach, including different microscopy techniques (SEM, TEM, fluorescence in situ hybridization and immunocytochemistry) as well as metagenomics techniques. We were able to identify the mineralogical composition of the layers that constitute the mineral deposit where the cyanobacteria are located. The layer associated with cyanobacteria is mainly composed of natrojarosite with a pH of 0.8, a common mineral precipitate given the ionic conditions of the Tinto basin [23]. The use of fluorescence in situ hybridization using specific probes has shown a strong association of cyanobacteria with a consortium of bacteria in the green-blue layer.

The identified cyanobacteria are characterized by having a cell envelope that is composed of a thick polysaccharide capsule, probably required to protect the cell from the extremely acidic conditions of the environment. The exterior is covered by a layer of oscillins (fibrous proteins) and finally, a mucous layer above the oscillin layer. The organelles present in these cells are carboxysomes, which are responsible for the fixation of CO_2_, and different types of granules holding possible compounds like glycogen, polyphosphate and lipids, ribosomes, and thylakoids. The nucleoid is surrounded by riboplasm, and the thylakoid system is found in the peripheral cytoplasm. These structures belong to typical cyanobacteria [40]. Genes related to some of these structures, e.g., glycogen debranching protein or polyphosphate kinase 1, have been identified by metagenomic shotgun, confirming the microscopy and inmunocytochemistry observations (Appendix A). 

To identify which cyanobacteria were involved in the microbial mat, many sequences belonging to this phylum and obtained from the green-blue layer of the stratified ecosystem were analyzed, highlighting the presence of members of several cyanobacterial families such as Synechococcaceae, Chroococcaceae, and Cyanobacteriaceae. Although, more in-depth molecular studies are needed to unambiguously identify the species present in the sample. The taxonomic assignment of open reading frames indicates that up to 1.5% of the aligned DNA reads belong to the Cyanobacteria clade, a value which agrees with the cyanobacteria cell number obtained by hybridization. The identification of genes coding for the Photosystem I iron–sulfur center protein PsaC, the Photosystem II D2 protein, the allophycocyanin subunit beta, several ribosomal proteins (S6 and L34), the PCC 6312 redox-regulated ATPase YchF, the PCC 71132Fe-2S iron–sulfur cluster binding domain, or the PCC 7428 peroxiredoxin, with high levels of coverage and identity similarity underlines the presence of different cyanobacterial genes in the system. 

Once the presence of cyanobacteria in the green-blue layer has been identified by microscopy and metagenomics, it was considered of interest to understand their ecological role in the microbial mat. An important ecological role reported for cyanobacteria is nitrogen fixation [41], especially in oligotrophic environments like Río Tinto. The process of nitrogen fixation is catalyzed by the enzyme nitrogenase [42]. Our metagenomic analysis indicates the presence of a nitrogen-cycle-related gene, e.g., ntcA gene, which encodes for a regulatory protein required for the expression of the genes repressed by ammonium in cyanobacteria [43], or the metal-binding protein WP_045870890.1, showing similarity with the Nostocales multispecies metal-binding protein. This apoprotein with two 4Fe-4S centers FA and FB of photosystem I (PSI) is essential for photochemical activity [43] (Appendix A). Even though we reported some sequences related to nitrogen fixation, deeper studies are needed to corroborate the ecological role of cyanobacteria in the extreme acidic environments of Río Tinto. 

Another important issue is to understand how these cyanobacteria can operate in the extreme conditions of the ecosystem. Little is known about the adaptation mechanisms of cyanobacteria to environments as adverse as the hyperacidic pH that we described in this work. An ATP-dependent DNA helicase has been reported to play a central role in the reaction driving the unwinding of the DNA helix and restoring DNA after damage [44]. We hypothesize that the cyanobacterial genomes are subject to damage by chemical agents in this hyperacidic environment with a high concentration of heavy metals. Thus, the identification of the ATP-dependent DNA helicase RecG (Appendix A) in the metagenome indicates its possible role in different DNA repair systems [45]. The presence of Thioredoxin TrxC (Appendix A), a protein involved in the regulation of redox balance in cells, could act as an antioxidant enzyme and plays a crucial role in multiple biological processes [46]. Thioredoxin TrxC (Appendix A) also has a role in metal binding, interacting selectively with metal ions, and it has been suggested that it could play a role in regulating photosynthetic adaptation to low carbon and/or high radiation conditions as well [47]. YchF is an unconventional member of the universally conserved GTPase family, which preferentially hydrolyzes ATP rather than GTP [48]. This ATPase has been associated with various cellular processes, including DNA repair. In particular, a possible role in regulating the oxidative stress and antioxidant response in bacteria has been suggested [49]. But the adaptation mechanism of cyanobacteria to survive in very acidic conditions is an open issue that requires further analysis. In any case, these metagenomic results strongly indicate that the identified genes are able to overcome a part of the problems produced by the extreme conditions in which the cyanobacteria develop. 

On the other hand, the analysis performed with VIS-NIR spectroscopy showed the presence of absorption peaks of 400–500 nm related to the charge transfer of iron for the orange mineral crust, and an absorption of around 430 nm for the white crust also related to the presence of natrojarosite [50]. In contrast, the spectrum collected on the green zone does not show a red slope in the visible range but does show a Chl-a absorption at 620 nm, a fluorescence peak at 650 nm, and a scattering peak in the green region (532 nm) consistent with the published cyanobacterial spectra [51,52]. A sample of the green layer was measured in the laboratory by micro-Raman spectroscopy. The collected spectrum showed Raman peaks at 1522, 1158, and 1106 cm^−1^, commonly attributed to the *ν1*(C=C), *ν2*(C-C) stretching mode and *δ*(C-CH_3_) bending mode of carotenoids [53,54]. Specifically, the Raman peak related to the *ν1*(C=C) vibration at 1522 cm^−1^ falls in the range of values found in the presence of lutein, β-carotene, zeaxanthin, and astaxanthin [54]. The spectroscopy data underlined the presence of iron minerals and the elements associated with photosynthesis in the analyzed mineral crusts in which cyanobacteria were detected.

Considering all the data generated using the different methodologies, we can claim the presence of cyanobacteria in a mineral layer composed of natrojarosite and at pH 0.8. This is the first time that the presence of cyanobacteria is reported in an environment below pH 1.

The differences between the radiation spectra of the surface and the interior of the biofilm indicate an attenuation of the internal radiative incidence. These differences seem to indicate greater stability within the ecosystem, in addition to an irradiation range compatible with photosynthetic active radiation (PAR), that is, compatible with greater photosynthetic efficiency. The filtration of the high incident radiation from this town in southern Spain generates a stabilization of the radiation within the ecosystem. This filtration is mainly due to the layers of salts precipitated on the surface of the biofilm. The upper mineral crust of the biofilm (Figure 1) is what generates this filtration that attenuates the radiation inside the biofilm.

This acidic, endolytic environment with cyanobacteria studied in this work has important astrobiological implications, since the search for life on Mars is the search for protected environments, such as mineral deposits, due to the extreme conditions existing in the surface of the planet [55]. In some areas of Mars, the presence of jarosite has been reported to be generated from the weathering of iron minerals [56] and the presence of acidic waters giving rise to mineral deposits in different areas of the planet [57,58]. Thus, future Mars missions should pay attention to the protective environments produced by iron minerals like jarosite in their search for signs of life. The recent demonstration of methane production by cyanobacteria under anaerobic conditions [59] gave an interesting connection between photosynthesis and the production of methane on Mars, a source of energy for methanotrophic microorganisms [60]. The fixation capacity of cyanobacteria for N_2_ and CO_2_, both components of the Mars atmosphere, makes cyanobacteria an interesting type of microorganism for the search of life on the red planet. Furthermore, the future development of human colonies on Mars will require the use of microorganisms able to make use of the sources of energy available, such us solar radiation, to produce different compounds of biotechnological interest, cyanobacteria again being an interesting possibility to explore [61]. The presence of cyanobacteria in the extremely acidic conditions existing in one of the best mineralogical and geochemical Mars analogues, Río Tinto [62], makes their discovery interesting in the search for signs of life on the red planet. Further research should concentrate on the isolation of these cyanobacteria or if they require the association with other bacteria, to study the consortium to gain information of its operation in the extreme conditions existing in Río Tinto and probably on Mars. 

The main implication of this work is the contribution to the description of an extreme ecosystem where we find species of cyanobacteria not previously described at such low pH values. Furthermore, in relation to the possible future human colonization of the planet Mars, we began the study of organisms that can support astronauts. However, the limitation identified is the lack of knowledge in this technical support, so it is necessary to carry out new research to better understand the way in which we can support the generation of biomass or the production of oxygen in these future human colonies.

## 5. Conclusions

In this work, the presence of cyanobacteria associated with an acid biofilm of below pH 1 has been reported. We have studied the ultrastructure of these microorganisms using scanning and transmission electron microscopies, with which the cellular structures that characterize these microorganisms have been studied. The ultrastructure and organelles that define cyanobacteria have been reported using immunocytochemistry. 

Shotgun metagenomics was carried out to characterize the microbial community present in the acidic microbial mat. Taxonomic assignment of the reads indicated that up to 0.46% of the classified sequences belong to cyanobacterial microorganisms. The Synechococcales and Nostocales orders stand out, followed by the Oscillatoriales and Chroococcales orders. In addition, other metabolic activities from cyanobacteria were identified using taxonomic assignment (up to 1.47% of aligned DNA reads belong to Cyanobacteria clade).

We can conclude that these species are present in association with biofilms at below pH 1, and the astrobiological, and the future use of these microorganisms in support of a possible human colonization of Mars implications have been depicted.

## Figures and Tables

**Figure 1 microorganisms-12-00829-f001:**
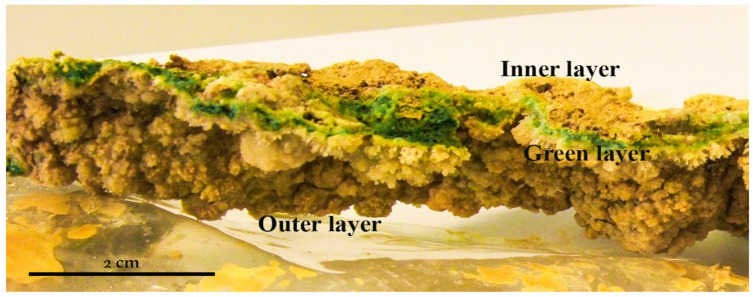
Microbial mat sample consisting of a mineral layer on the surface, with a blue-green layer in the middle and finally, a brown ocher layer at the bottom.

**Figure 2 microorganisms-12-00829-f002:**
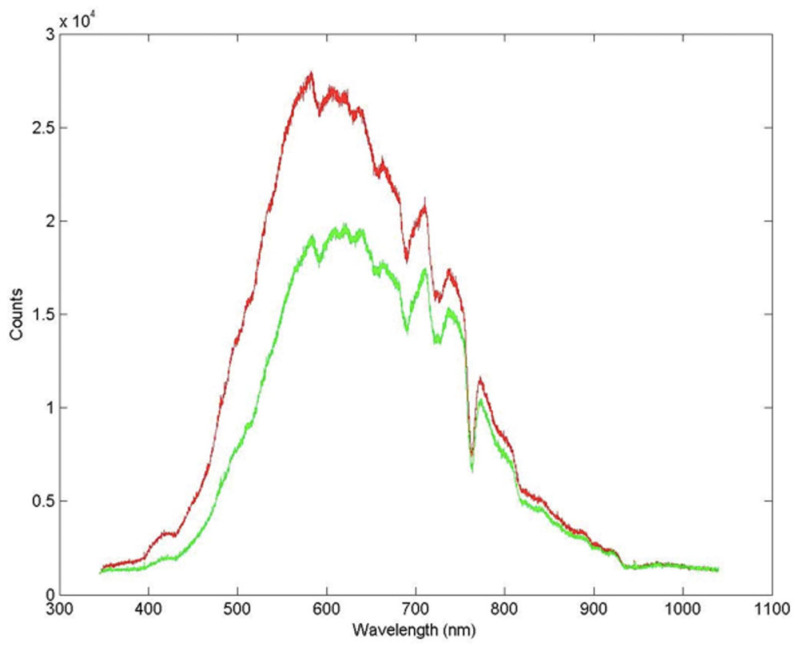
Radiation spectrum at the surface (red) and inside the microbial mat (green).

**Figure 3 microorganisms-12-00829-f003:**
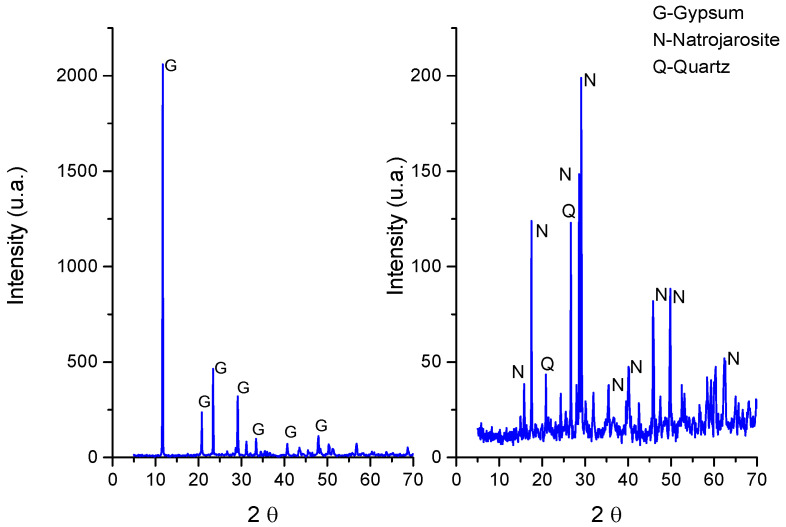
XRD analysis of the mineral layers. On the left is the outer layer. On the right is the brownish inner layer.

**Figure 4 microorganisms-12-00829-f004:**
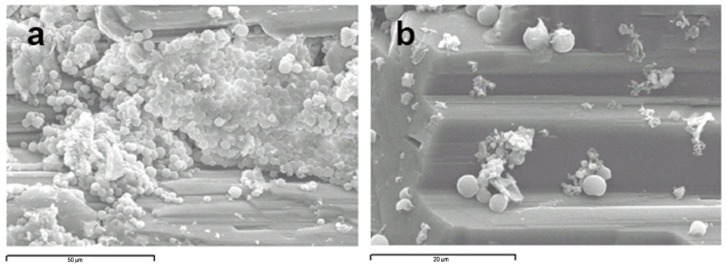
(**a**) Scanning electron microscopy images of the green-blue layer in which spherical morphologies with some surface features are observed; (**b**) close-up of some individual cells attached to the mineral surface.

**Figure 5 microorganisms-12-00829-f005:**
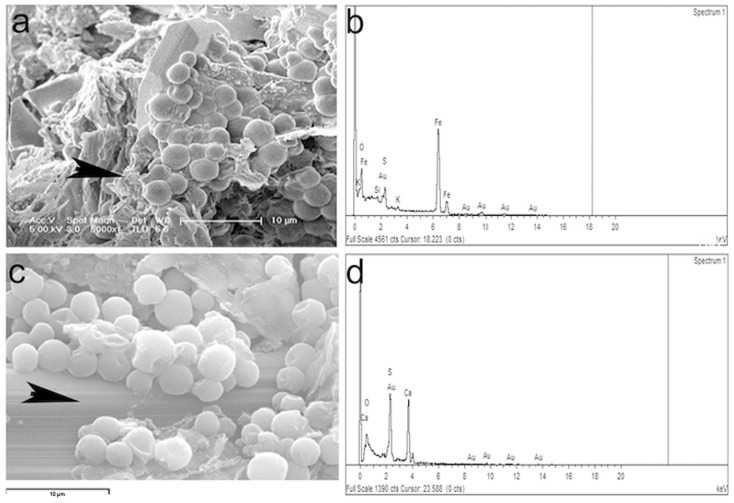
EDX analysis in two spots (arrowheads) of the green-blue layer. Location of the spot (**a**) for the EDX analysis (**b**). With more zoom another analysis locations was selected (**c**) for the EDX analysis in (**d**).

**Figure 6 microorganisms-12-00829-f006:**
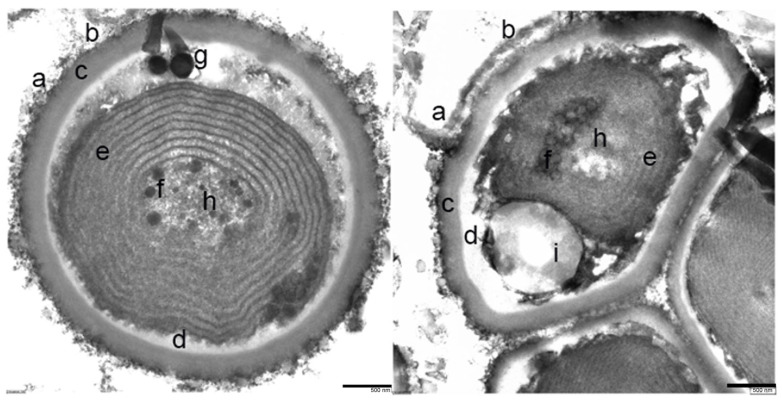
Cells from the green-blue layer observed by TEM. Left image: individual spherical cell morphology. Right image: consortium of multicellular morphologies. (a) Fibrous protein and mucilaginous layer outside the cell envelope; (b,c) polysaccharide capsule; (d) cell membranes; (e) thylakoid membrane system; (f) putative carboxy-type corpuscles; (h) granules of variable density that may contain glycogen; (g) polyphosphates; and (i) lipid drops.

**Figure 7 microorganisms-12-00829-f007:**
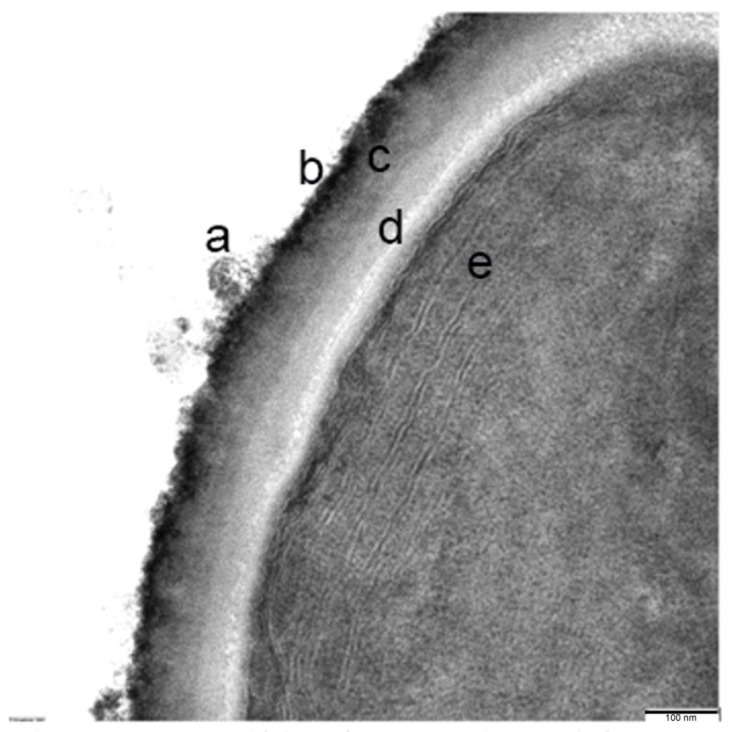
Detailed close-up section of the membrane: (a) fibrous protein and mucilaginous layer outside of the cell envelope; (b,c) polysaccharide capsule; (d) cell membrane; and (e) thylakoid membrane system.

**Figure 8 microorganisms-12-00829-f008:**
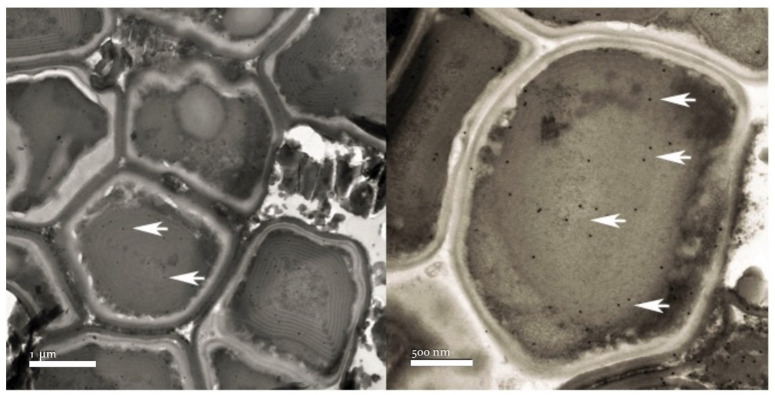
α-major carboxysome shell protein immunolocalization in the cyanobacteria cells indicated by the arrows.

**Figure 9 microorganisms-12-00829-f009:**
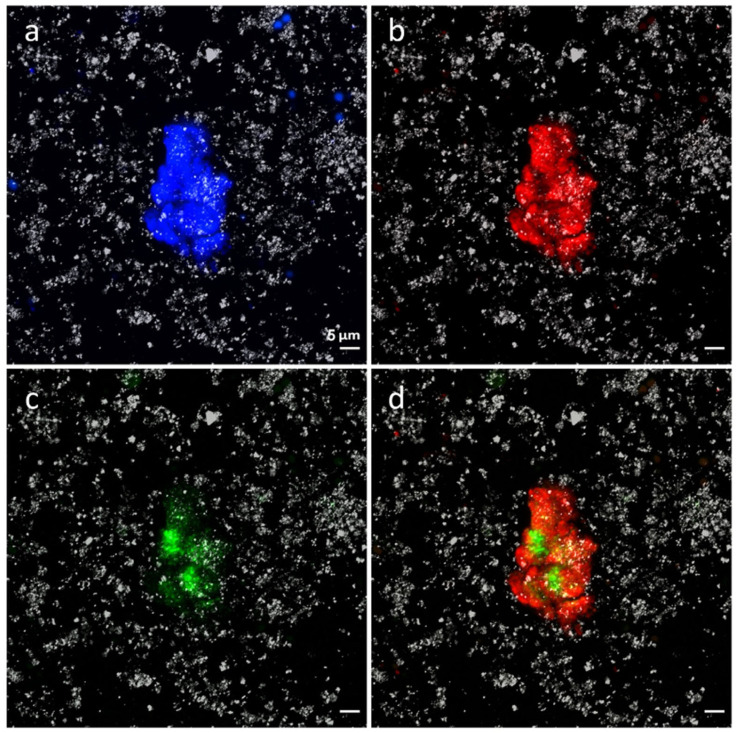
Cyanobacteria and Bacteria detected by CARD-FISH. (**a**): DAPI general stain; (**b**): bacteria detected by EUB338 I–III probe (red); (**c**): cyanobacteria detected by CYA 361 (green); (**d**): merge of (**b**,**c**) images.

**Figure 10 microorganisms-12-00829-f010:**
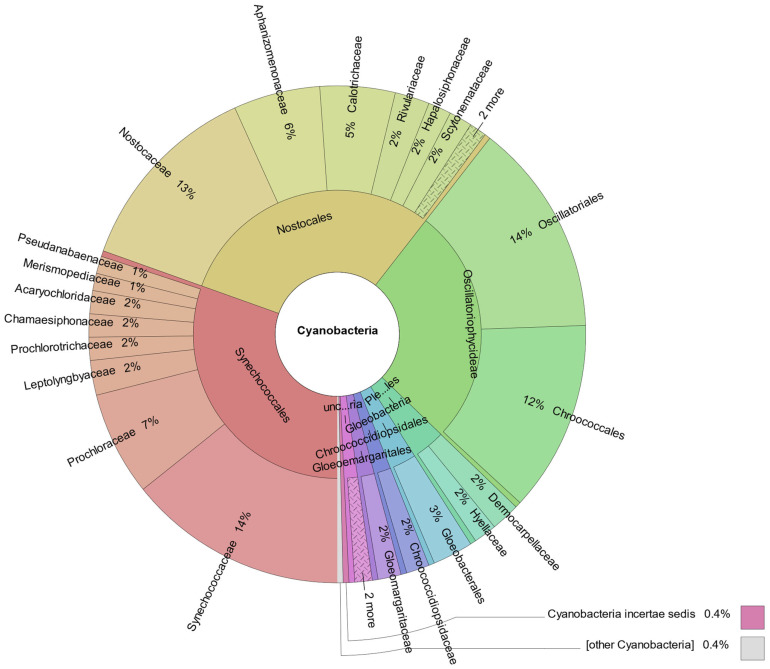
Taxonomic distribution of shotgun metagenomic reads assigned to Cyanobacteria phylum.

**Figure 11 microorganisms-12-00829-f011:**
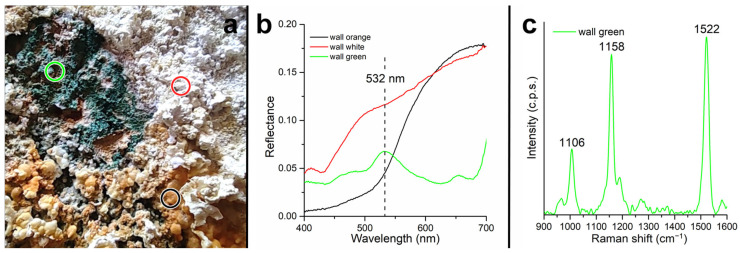
Microbial mats spectral analysis. (**a**) View of the microbial mats, the colored circles indicate the points where the spectra were collected. The color of the circles corresponds to the color of the spectra in the panels (**b**,**c**). (**b**) Visible spectra collected in situ. (**c**) Micro-Raman spectrum collected in the laboratory on a sample of the green area.

## Data Availability

The cyanobacteria related contigs sequences analyzed in this study were submitted to the European Nucleotide Archive (ENA) under the accession number ERZ1300892.

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
