# Peer review of "Association of Acidotolerant Cyanobacteria to Microbial Mats below pH 1 in Acidic Mineral Precipitates in Río Tinto River in Spain"

_microorganisms, 2024, doi:10.3390/microorganisms12040829_

Round 1

Reviewer 1 Report (Previous Reviewer 1)

Comments and Suggestions for Authors

The manuscript by Gomez et al has improved considerably with respect to its previous submission. The authors successfully addressed all my previous comments. I just have a small comment that should be address before its publication.

 Line 89: Please add more information regarding how the samples was fixed in the field. Which fixative was used? At which concentration?

Author Response

First of all I would like to thanks to the reviewers for their help on the improvement of the present manuscript.
We´ve accomplished all the reviewers comments.
The answers to the comments are:

Reviewer 1:
Line 89: Please add more information regarding how the samples was fixed in the field. Which fixative was used? At which concentration?

Authors´ Answer: Samples for culturing and direct optical and electronic microscopies observation were not fixed in the field. They were stored and maintained at 4ºC during transportation to the lab. Samples for metagenomics were frozen in dry ice.
The samples for CARD-FISH were fixed immediately in the field, with 4% formaldehyde in phosphate-buffered saline (PBS) for 2 h at room temperature and stored at 4 °C until further processing. Once in the lab, samples for SEM were fixed with 4% formaldehyde in phosphate-buffered saline (PBS) for 2 h at room temperature and stored at 4 °C until further processing. The samples for TEM were fixed in 4% paraformaldehyde and 2% glutaraldehyde, in phosphate buffer pH 7.4, for two hours at RT under stirring and overnight at 4ºC, followed by extensive washing in the same buffer.

Reviewer 2 Report (New Reviewer)

Comments and Suggestions for Authors

Overall, this study is interesting and has some novelty, however the reviewer has provided several comments to improve overall quality of the manuscript.

Comments:

1.      In the title, change “Río Tinto (Spain)” to “Río Tinto River in Spain”.

2.      Abstract: At the end, include a statement on the important implications of this work. Also, since it is a research-based study, it would be good to include the key quantitative data.

3.      Line 50- 51: “Río Tinto has an average pH value of 2.3, with several sampling stations showing a pH below 1”. Could you provide more information about why the river is extremely acidic, also whether the pH of the water is acidic throughout the year.

4.      Are there any studies on the detailed characterization of water qualities including presence of metals/heavy metals.  

5.      Line 69: Change “In this research is reported” to “In this research, it is reported”.

6.      Line 81: “The sampling campaign was run in May 2010 for 5 days.”. The samples were collected nearly 14 years back, hope the sample integrity is not compromised specifically for the microbial analysis. Genomic DNA may degrade over time with improper storage and handling. For five days sampling, is it continuous five days as well as are there any weather changes in these five days. Need to mention how the collected samples were stored (temperature) until analysis were done.

7.      Line 163: “2.7. CARD-FISH”. Give the full name of the abbreviation.

8.      Line 105 – 162: The reviewer is wondering whether any pre-treatments (e.g., removal of moisture by drying) were done with the sample for physicochemical characterization by different techniques such as XRD, SEM, TEM, etc.

9.      Figure 2: Explain more about the difference radiation spectra between surface and inside microbial mat. Is it mainly due to differences in the microbial composition?

10.  Figures 6 – 7: The reviewer is suggesting to add the color images of the TEM analysis if available. It will improve better visualization of cell components specifically Figure 7.   

11.  At the end of Discussion, add a section describing the important implications and limitations of this work.

12.  The “Conclusions” section is missing, thus needs to add an individual section on Conclusions by highlighting the key results and observations found in this work.

Comments on the Quality of English Language

Careful checking of typographical and grammatical errors is suggested. 

Author Response

First of all I would like to thanks to the reviewers for their help on the improvement of the present manuscript.
We´ve accomplished all the reviewers comments.
The answers to the comments are:

Reviewer 2:
Comments:
1.In the title, change “Río Tinto (Spain)” to “Río Tinto River in Spain”.

Authors: Done

2.      Abstract: At the end, include a statement on the important implications of this work. Also, since it is a research-based study, it would be good to include the key quantitative data.

Authors: Done. We have included quantitative data in the new version. Up to 0.46% of the classified sequences belong to cyanobacterial microorganisms and 1.47% of aligned DNA reads belong to Cyanobacteria clade, has been included.
We think that the important implications of this work is already included in the abstract with the sentence “demonstrating, for the first time, the existence of acid tolerant cyanobacteria in a hyperacid environment with pH bellow 1”.

3.      Line 50- 51: “Río Tinto has an average pH value of 2.3, with several sampling stations showing a pH below 1”. Could you provide more information about why the river is extremely acidic, also whether the pH of the water is acidic throughout the year.

Authors: Done. The sentence 

“The presence of iron- and sulfur-oxidizing and reducing prokaryotes and its interaction with the Iberian pyritic belt located in the subsurface are responsible for the extreme conditions of acidity (mean pH 2.3) and the high concentration of heavy metals found in its waters (Fe, Cu, Zn, As, Cr).” 

has been included in the new version.

4.      Are there any studies on the detailed characterization of water qualities including presence of metals/heavy metals.  

Authors: Yes, we have been studying the Rio Tinto ecosystem from more than 25 year ago and we have been measuring regularly all the physico-chemical parameters. There are several publications with data about Rio Tinto water content and the concentration of different metals and heavy metals present in its waters, as for example ref. 21, and the other Rio Tinto related references in this work.

5.      Line 69: Change “In this research is reported” to “In this research, it is reported”.
Authors: Done

6.      Line 81: “The sampling campaign was run in May 2010 for 5 days.”. The samples were collected nearly 14 years back, hope the sample integrity is not compromised specifically for the microbial analysis. Genomic DNA may degrade over time with improper storage and handling. For five days sampling, is it continuous five days as well as are there any weather changes in these five days. Need to mention how the collected samples were stored (temperature) until analysis were done.

Authors: Most of the lab work was done 14 years ago, just after sample arrived to the lab. In any case, all samples were maintained refrigerated or under frost conditions. 
Regarding weather conditions, Rio Tinto is located at Huelva province, south-west of Iberian peninsula. The weather conditions at the south of Spain, including Rio Tinto location, is very stable. During that five days there were no significative weather changes. 

7.      Line 163: “2.7. CARD-FISH”. Give the full name of the abbreviation.

Authors: Done.

8.      Line 105 – 162: The reviewer is wondering whether any pre-treatments (e.g., removal of moisture by drying) were done with the sample for physicochemical characterization by different techniques such as XRD, SEM, TEM, etc.

Authors: Samples for culturing and direct optical and electronic microscopies observation were not pretreated in the field. It was stored and maintained at 4ºC during transportation to the lab. Samples for metagenomics were frozen in dry ice. The samples for CARD-FISH were fixed in the field as explained in the text, immediately. Samples for CARD-FISH were fixed with 4% formaldehyde in phosphate-buffered saline (PBS) for 2 h at room temperature and stored at 4 °C until further processing. Once in the lab, samples for SEM were fixed with 4% formaldehyde in phosphate-buffered saline (PBS) for 2 h at room temperature and stored at 4 °C until further processing. In the case of TEM, samples were fixed in 4% paraformaldehyde and 2% glutaraldehyde, in phosphate buffer pH 7.4, for two hours at RT under stirring and overnight at 4ºC, followed by extensive washing in the same buffer.

9.      Figure 2: Explain more about the difference radiation spectra between surface and inside microbial mat. Is it mainly due to differences in the microbial composition?

Authors: Done. The paragraph “The differences between the radiation spectra of the surface and the interior of the biofilm indicate an attenuation of the radiative incidence inside. These differences seem to indicate greater stability within the ecosystem, in addition to an irradiation range compatible with photosynthetic active radiation (PAR), that is, compatible with greater photosynthetic efficiency. The filtration of the high incident radiation from this town in southern Spain generates a stabilization of the radiation within the ecosystem. This filtration is mainly due to the layers of salts precipitated on the surface of the biofilm. The upper mineral crust of the biofilm (fig. 1) is what generates this filtration that attenuates the radiation inside the biofilm.”
has been included in the new version.

10.  Figures 6 – 7: The reviewer is suggesting to add the color images of the TEM analysis if available. It will improve better visualization of cell components specifically Figure 7.   
Authors: We don´t have TEM color images.

11.  At the end of Discussion, add a section describing the important implications and limitations of this work.

Authors: Done. The following paragraph has been added to the new version of the draft: “The main implication of this work is the contribution to the description of an extreme ecosystem where we find species of cyanobacteria not previously described at such low pH values. Furthermore, in relation to the possible future human colonization of the planet Mars, we began the study of organisms that can support astronauts. However, the limitation identified is the lack of knowledge in this technical support, so it is necessary to carry out new research to better understand the way in which we can support the generation of biomass or the production of oxygen in these future human colonies.”

12.  The “Conclusions” section is missing, thus needs to add an individual section on Conclusions by highlighting the key results and observations found in this work.

Authors: Done.

Round 2

Reviewer 2 Report (New Reviewer)

Comments and Suggestions for Authors

The revised manuscript can be accepted. 

Comments on the Quality of English Language

Minor English editing is recommended. 

This manuscript is a resubmission of an earlier submission. The following is a list of the peer review reports and author responses from that submission.

Round 1

Reviewer 1 Report

Comments and Suggestions for Authors

The manuscript by Gomez et al., studied the presence of acidophilic cyanobacteria in extremely acidic ecosystems (pH below 1). By using molecular approaches, the authors described for the first time, cyanobacteria in these types of environments. Although these results will contribute to better understand cyanobacteria distribution in the environment, and therefore being of potential interest for a broad audience, several improvements must be done before the acceptance of this manuscript.

Comments:

Title: the expression “at below pH 1” sounds odd. I would change it to “at pH below 1”. However, I would suggest to modify the title to “Association of acidophilic cyanobacteria with microbial mats in acidic mineral precipitates below pH 1 in Río Tinto”

Abstract: The abstract should be improved to really described what was performed in this manuscript.

Lines 30-32: Also include hot springs and add references

Line 46: 2 in CO2 should be subscript

Lines 88-89: which type of precipitates? carbon precipitates? iron precipitates? please also add references

Lines 85-97: Information regarding the conditions at which the samples were stored back in the laboratory are missing. Why samples where not stored at cold temperatures in the field? For molecular analysis samples are usually frozen in the field with either dry ice or liquid nitrogen and then stored at -20C in the lab.

Line 124: which samples were used? Mineral deposits? Sediments? Water? Please be more specific

Lines 124-135: Why samples were not fixed?

Lines 136-159: Where thin cut performed? or the authors observed whole cells by TEM

Line 137: which samples were used? Mineral deposits? Sediments? Water? Please be more specific

Line 137: add a space between 0.1 and M. Keep consistency through out the text. For example, in line 144 there should also be a space between 50 and mM

Line 150: it is not necessary to mention bovine serum albumin again, because it was defined in line 147. Use just BSA instead

Line 152: Add a space between Tween and 20

Line 156: Add a space between 1 and h. Please carefully check throughout the manuscript these types of inconsistencies and fix accordingly

Line 157: Add a space between Tween and 20. Please carefully check throughout the manuscript these types of inconsistencies and fix accordingly

Line 161: Add concentration of PBS buffer

Line 163: black membrane filters. Add more information about the washes for example which concentration of PBS buffer was used, how many times was washed, the last step was with just absolute ethanol?

Lines 168: Add space between 80 and mM

Line 214: maybe the authors wanted to say sample instead of ample?

Lines 213-214: The information in these lines is repetitive, please fix.

Lines 218-224: Why this measurement was performed at noon? In lines 203-204 the authors said that the minerals are present mostly on the shade side of the walls with very few hours of direct light. Was the radiation measurement done when the minerals were exposed to direct light? The authors said that the incident radiation over the five days of measurement was the same at the same time of the day, and therefore noon measurements were chosen as representative. However, since it is not clear if at noon the mineral was exposed to direct light or not, it is hard to validate that noon measurements are representative. Moreover, since most of the day these minerals crusts are not exposed to direct light, it would have been ideal to do the measurements at two time points, ie, with and without direct light. Why measurements were performed at one point only?

Figure 2: The label of the Y axe is not clear, please fix. Also remove “spectrum comparative” from above the figure.  

Figure 4 and 5: Based on these figures, the images shown are ultrathin section observed by TEM. However, the authors never mentioned the utilization of a microtome or any step for the preparation of the thin section. TEM can also be used to see cells but to see the inner parts and layers such as cell wall or cell membrane a thin cut must be done. Please clarify in the text and in the figures. Moreover, ultra-thin cuts observed by TEM is not an ideal technique to determine cell morphologies and sizes. It is better to use SEM or TEM without thin section. Fix accordingly.  Add information about the scale bar because it is not possible to read it from the figure.

Lines 250-251: The authors said, “a cell wall that is composed of a plasma membrane and an outer membrane, with a layer of peptidoglycan of varying density between them”. However, the plasma membrane it is not part of the cell wall. The authors should say “a cell envelope” instead of “cell wall”

Line 259: Based on what is observed from figure 6, the authors cannot say that this is a cyanobacteria-containing biofilm. This taxonomical classification can not be determined by SEM.

Lines 271-275: It is not clear how these techniques can validate the presence of “live” cyanobacteria. Although CARDFISH and metagenomics approaches are useful to show the presence of a certain group of microorganisms (cyanobacteria in this case), these techniques do not provide information about the viability of the microorganisms.

Figure legend 6 and 7: I would not say cyanobacterial biofilm since there is no information (at least until this point) to prove that. Moreover, although in the next section the authors described the presence of cyanobacteria in the biofilm, the abundance of this group was just 1% so I would not consider this biofilm as a “cyanobacterial biofilm”. Cyanobacterial containing biofilm is more appropriate as a term but should be mentioned after the molecular identification of this group.

Figure 8: I would change the color of the label from black to white because it is hard to see the legend in the figure. For example, I can not read the number above the bar in the left figure.

Figure 10: If possible, i would try to increase the font in the figure.

Lines 319-320: 95.1% coverage and similarity of 98.8%.

Line 322: 58.2% and identity percentage of 84.3%.

Line 324: 78.8% identity

Figure 11: Fix label b and c in figures

Lines 341-342: The authors mentioned “white, yellow and green” but before the authors referred to white, orange (and not yellow) and green.

Line 386: “cell envelope” instead of “cell wall”

Lines 386-387: Please add references

Line 444: Genus and specie must be italicized

Lines 452-453: Can the authors provide more information regarding how their results indicate that the cyanobacteria present in the environment are growing and active? I did not see results that support that

Reviewer 2 Report

Comments and Suggestions for Authors

The manuscript submitted by Gomez et al. is a significant work in the extreme biology. The detailed information on the physic-chemical parameters of the samples with unculturable microbial diversity is a good contribution in order to understand the biological processes in extreme hydrogen ion conditions, at some extent. However, I have some minor comments:

1. Please provide a table of physiological parameters (temperature, pH etc.) with the standard error bars, that would clear more about the range of the parameters investigated. I asked because the range of pH is the most important ecological factor in this study. 

2. Author's have analyzed the cyanobacterial community by shotgun metagenomics. The functional attributes needs more discussion towards its possible involvement in adaptations at environmental pH conditions. 

3. I would suggest to improve the discussion on stress related parameters (with updated reference 2022-2023, if possible) which will strengthen the importance of findings in the astro-biological aspects.

I recommend this work to be published in the reputed journal Microorganisms after considering the provided comments.

Thanks

Reviewer 3 Report

Comments and Suggestions for Authors

I undertook to review this manuscript with great enthusiasm, since acidophilic cyanobacteria are, indeed, a unique object.

To begin with, I was confused by the abstract, in which there was no specifics about the results obtained. I highly recommend correcting the abstract in such a way as to emphasize the specific achievements of this work.

However, when I read the article, I realized that such an abstract is not accidental. The work contains a lot of various information, but does not have a clear and precise statement of the problem. What did the authors want to describe? Acidophilic cyanobacteria? But then, having the whole arsenal of techniques used in the manuscript, they would have to taxonomically identify the organism they describe. This has not been done even as a first approximation. The results of molecular genetic analysis give a very general idea of the composition of cyanobacteria in the prokaryotic community, but are not correlated with the data of TEM, SEM, FISH... Moreover, strictly speaking, the authors are not allowed to designate microorganisms as acidophilic only by the fact of their observation under acidic conditions. They can have very strong defense mechanisms against acidic pH and be acid tolerant. Therefore, the acidophilicity of cyanobacteria must be proved. To do this, you need to isolate strains and find their growth optima. If this is not possible for any reason, then the authors should remove the definition "acidophilic" from the title and from the text.

Further, the purpose of the authors could be to show how mineral crusts with natrojarosite are formed with the participation of cyanobacteria. In this case, it is necessary to give more information about mineral precipitation under acidic conditions and about the role of microorganisms in this process. In addition, the role of cyanobacteria in this will have to be proved, especially since the authors write that cyanobacteria make up about 0.5% of the entire taxonomic diversity of the community. The entire community, not just its cyanobacterial part, must be analyzed. The potential of natrojarosite precipitation by other microorganisms in the community must be considered.

In general, based on the data presented in the manuscript, I see the potential for writing two articles, each of which will have its own clear statement of the problem. I would recommend the authors to share the material. In its current form, I do not consider it suitable for publication. Authors can defend their point of view and leave all the data in one article, but then they need to set goals and objectives very clearly. Now everything is blurry, which makes the scientific value of the article low.

Below are some specific remarks that I had while reading the article.

Lines 79-83: There is no need to describe the results here, especially without specifics. Here it is necessary to set clear goals and objectives.

Lines 87-89: Is washing with water a periodic (repeating) phenomenon? Are the crusts not constantly in the water flow? If so, is it possible to speak further about the constancy of conditions?

Lines 137-138: "Samples were embedded in PB buffer (0.1M sodium phosphate pH 7.4) at room temperature for 2 hours under stirring" - For what? the cells were not yet fixed, most likely it was a strong stress for them if they are adapted to a pH below 2.

Lines 139-140: "were fixed in 4% paraformaldehyde and 2% glutaraldehyde, pH7,4" - It is a bit much, but ok, it's unlikely to hurt much.

Point 2.6. Dehydration steps and the types of the resin where the samples were enclosed for TEM and immunocytochemistry are not described. What was the contrast for the study of the ultrastructure? Did you use osmium for postfixation?

Lines 150-156: "Grids were floated on 15 mL of diluted anti α-major carboxysome shell protein, 1:2, 1:20 or 1:100 in TBS (30 mM Tris–HCl pH 8.2, 150 Mm NaCl, 0.02% azide) containing Tween 20 with 20 mg of bovine serum albumin (BSA) per ml and incubated for 1 h at 37 °C. Grids were incubated with the primary antibody (α-CsoS1 dil 1/2, Agrisera) in 5% BSA in T / TBS, for 90 min at 37 °C" - I didn’t understand anything. "Anti α-major carboxysome shell protein" - what is it? An antibody? But then the you write about primary antibodies to α-CsoS1, which is carboxysome shell protein. How many variants of the first antibodies were there? "dil 1/2" -  is a dilution? It seems too little.... Agrisera antibody ID not specified.

Lines 153-157: Information about the second antibody is written twice: "Incubations with goat antirabbit IgG-gold conjugate (GAR 15 nm, Biocell, Cardiff, UK), diluted 1:40 in TBST (Tris-buffered saline with Tween 20) with 2 mg of BSA per mL , were carried out for 1 h at 37 °C. Incubation with the colloidal gold conjugated secondary antibody (PAG15 dil 1/50)) in 5% BSA in T / TBS was carried out for 1h at 37 °C and finally slides were washed with TBS + 0.2% BSA + 0.1% Tween20, 3 x 5 min at room temperature". (Two different second antibodies? Unclear. Please, specify.

Lines 203-205: "where there is thermal 204 and pH stability"  Where is the measurement data? The statement is not supported by evidence at this point in the text.

Fig. 1. А scale bar or some object in the photo to understand the scale is needed.

Figs 4-5. I cannot agree with the authors' interpretation of cellular structures. The plasma and outer membranes are membranes, they have a clear structure (like thylakoid membranes). I do not see these characteristic structures in the photo. The murein layer should be seen as electron dense layer but not as electron transparent. I have a feeling that the cells were destroyed in the process of sample preparation, especially the cell membranes. It seems that the cells shrank and moved away from the cell walls. Moreover, the designations b, c, d in Fig. 4 and Fig. 5 do not coincide with each other. To my mind, on fig. 4 designation a is most likely a released polysaccharides, b - a thick polysaccharide capsule, but the membranes seem to have been destroyed specifically and they are simply not visible (the same for fig. 5). In fig. 5 both membranes should be directly under the letter d. So I would say that these cyanobacteria have a very dense layered polysaccharide capsule. And the cell membranes are poorly preserved or not visible with this quality of photos (but the first one is more likely). Is it possible to take pictures under a light microscope? Such a capsule will definitely be noticeable.
In addition, the designation of structures carboxysomes in Fig. 4b is doubtful. There is something structured, but it is unambiguously difficult to interpret - you need a photo with high magnification and resolution. Polyphosphates are probable, but without the elemental composition, this can only be assumed. Cyanophycin and lipids, in the absence of elemental analysis, are also better labeled in a hypothetical manner. Glycogen raises questions because it is usually deposited between the thylakoids. The gas vacuoles mentioned in the text are not shown in the photo.

Line 256: Fig 5 does not correspond to the actual phrase in the text.

Lines 276-280: It is not clear why the Authors did immunolocalization with carboxysomes? Why is this analysis necessary? In this case, these data seem to be excessive. But, perhaps, a clear statement of the goal and objectives of the manuscript will answer this question.
In addition, typical carboxymes are again not visible, as in Fig. 3-5, and gold is scattered throughout the cell, which does not agree with the location of the putative carboxysomes in Fig. 4a. This needs some explanation.

Lines 289-292: How was the number of cells calculated? I didn't see this information in the materials and methods. It seems to me very difficult to count the number of cells inside mineral crusts.

Section 3.3. How do these data agree with the TEM and SEM data presented above, where only unicellular cyanobacteria are visible. This requires an explanation.

Lines 307-329. Redundant information that has nothing to do with anything. For what???

Section 3.4. The whole section also seems redundant. Why is this information needed? What new and important it proves? It is not clear why it was impossible to simply take the spectrum of the extracts (acetone or methanol, or another).

Section 4. Written very badly. In fact, it does not contain a discussion of the presented results, but is a text about everything and nothing.